# Scattering of a Bessel Pincer Light-Sheet Beam on a Charged Particle at Arbitrary Size

**DOI:** 10.3390/mi15080975

**Published:** 2024-07-29

**Authors:** Shu Zhang, Shiguo Chen, Qun Wei, Renxian Li, Bing Wei, Ningning Song

**Affiliations:** 1School of Physics, Xidian University, Xi’an 710071, China; qunwei@xidian.edu.cn (Q.W.); rxli@mail.xidian.edu.cn (R.L.); bwei@xidian.edu.cn (B.W.); nnsong@stu.xidian.edu.cn (N.S.); 2Key Laboratory of Optoelectronic Information Perception in Complex Environment, Ministry of Education, Xidian University, Xi’an 710071, China

**Keywords:** Bessel pincer light-sheet beam, charged particle, GLMT

## Abstract

Electromagnetic scattering is a routine tool for rapid, non-contact characterization of particle media. In previous work, the interaction targets of scattering intensity, scattering efficiency, and extinction efficiency of Bessel pincer light-sheet beams were all aimed at dielectric spheres. However, most particles in nature are charged. Considering the boundary condition on a charged sphere, the beam shape coefficients (BSCs) ($p_{mn},q_{mn}$) of the charged spherical particle illuminated by a Bessel pincer light-sheet beam are obtained. The extinction, scattering, and absorption efficiencies are derived under the generalized Lorenz–Mie theory (GLMT) framework. This study reveals the significant differences in scattering characteristics of Bessel pincer light-sheet beams on a charged particle compared to traditional beams. The simulations show a few apparent differences in the far-field scattering intensity and efficiencies between charged and natural spheres under the influence of dimensionless size parameters. As dimensionless parameters increase, the difference between the charged and neutral spheres decreases. The effects of refractive index and beam parameters on scattering, extinction, and absorption coefficients are different but tend to converge with increasing dimensionless parameters. When applied to charged spheres with different refractive indices, the scattering, extinction, and absorption efficiencies of Bessel pincer light-sheet beams change with variations in surface charge. However, once the surface charge reaches saturation, these efficiencies become stable. This study is significant for understanding optical manipulation and super-resolution imaging in single-molecule microbiology.

## 1. Introduction

Numerous particles in nature are positively charged, such as water droplets formed from sea spray, raindrops, and ice crystals during thunderstorms [1,2,3]; snowflakes; and electrically charged dust particles [1,4,5,6,7] in the upper atmosphere [8,9]. These particles can be charged [10], and people also believe that interstellar particles are charged [11,12]. Optical methods widely used in the microphysics of solid components [13] and chemical diagnosis [14] all rely fundamentally on the principles of electromagnetic interactions with small particles [15,16], namely, the measurement and analysis of scattered light.

The study of charged particle scattering characteristics began with research on neutral particles. Initially, Lorenz [17] proposed the scattering characteristic schemes for neutral particles, which Mie [11,18] further refined. Conversely, the concept of charged spheres was first theoretically discussed by Bohren and Hunt [19,20], and later improved upon by Klačka and Kocifaj [21]. Subsequently, Ilya E. Kuznetsov [22] and colleagues analyzed how electromagnetic wave scattering affects electrical inhomogeneities in precipitation, focusing on the resonant effects of charged aerosol particles on electrically active regions of clouds and precipitation. Similarly, Miroslav Kocifaj and colleagues [23] employed Discrete Dipole Approximation (DDA) to study the optical characteristics of plane waves incident on charged spheres, focusing on scattering parameters and resonance peak variations. However, DDA is limited to the particles whose size is close to the wavelength. Extending this research, Etai Rosenkrantz and colleagues [24] extended Klačka and Kocifaj’s [21] research, investigating the extinction, scattering, and absorption efficiencies of charged nanometer particles and studying plane wave scattering on non-spherical charged particles. Additionally, H.-Y. Li et al. [25] studied the scattering characteristics of plane waves/Gaussian beams on charged multiple spheres using GLMT theory. Notably, the most significant advantage of the GLMT [19,26,27] method is its efficiency and accuracy in handling scattering problems for spherical particles. This advantage is particularly relevant in Terahertz (THz) radiation. THz [28,29] beams are generated using various techniques, including photoconductive antennas [30] for broadband pulses, optical rectification [31] for high-intensity generation, and quantum cascade lasers [32,33] for continuous output. Due to their ability to penetrate non-metallic materials [34,35], non-ionizing nature [36], unique spectroscopic properties [37,38], high sensitivity to water content [39,40], and low scattering in complex media, Thz radiation is superior to visible light for many applications [41,42,43,44,45]. These attributes drive the preference for THz radiation in security screening [46,47], medical imaging [48,49], material characterization [50], and high-speed communications [39,40], offering safer, more detailed, and versatile solutions. Nonetheless, most of the time, the excitation sources for these scattering studies are primarily plane waves [51].

Exploring structured light beams [52] could lead to significant breakthroughs in optical communications, optical manipulation technology, microscopic imaging, and nanofabrication. Various types of classically structured beams have been studied. For example, the Bessel beam [53,54,55,56], known for its characteristics of no diffraction, self-restoration, and self-reconstruction [57], is stronger than the Bessel–Gaussian beam [58] and more stable than the Airy beam [59]. Previous research on light scattering and mechanical effects focused on neutral spherical particles, while studies on charged particles emphasized the Vector Bessel beam. Notably, Shuhong Gong et al. [60] and RX Li et al. [61,62,63,64] explored Vector Bessel beams scattering on charged spherical targets and the optical force effects with varying polarization modes. Furthermore, Yiming Yang et al. [65] examined the scattering distribution of charged balls under the influence of Vector Bessel beams. However, limited diffraction characteristics constrain the standard 3D Bessel beam and Bessel–Gaussian beam, while a weak bending angle restricts the Airy beam. In contrast, the Bessel pincer light-sheet beam [66,67] offers a significant bending angle and self-focusing capability, surpassing typical structured beams’ limitations. The interaction of the Bessel pincer light-sheet beam with a charged spherical particle extends the traditional scattering theory, leverages the unique properties of the beams, and has significant implications for various scientific and technological applications. This research offers more profound insights into the behavior of charged particles and drives advancements in environmental monitoring, medical imaging, and nanotechnology.

Building upon previous research [68,69], the aim of this study delves into the scattering characteristics of Bessel pincer light-sheet beams interacting with charged spheres. Utilizing the classical Mie coefficient of the charged sphere, the study derives the scattering field strength of the Bessel pincer light-sheet beam interacting with the charged sphere in Section 2. The analysis focuses on parameters such as scattering intensity, scattering efficiency, extinction efficiency, and absorption efficiency, with particular attention to the influence of beam parameters of the Bessel pincer light-sheet beam (such as scale parameter $\alpha_{0}$ and beam order) on the charged sphere (dimensionless size parameter kasize and particle refractive index), given in Section 3. The summary of this work and some perspectives are given in Section 4.

## 2. Methods

The angular spectrum expression of Bessel pincer light-sheet beam (according to the angular expression in [67,68]) is
(1)$$S_{m}\left(p,q\right)=\frac{kE_{0}}{\pi \alpha_{0}^{l}\sqrt{\alpha^{2}-p^{2}}}ℜ\left\{{\left(p+i\sqrt{\alpha_{0}^{2}-p^{2}}\right)}^{l}\right\}$$
where $\alpha_{0}$ is the beam scaling parameter of Bessel pincer light-sheet beam, ł is the beam order of Bessel pincer light-sheet beam, (p,q) are the directional cosines (p=sinα, q=cosα), and α is the angle of propagation of the individual plane wave. $E_{0}$ is the electric field amplitude. *k* is the wave number of the incident beam. *ℜ* is the real part of a complex number. *m* is the order of the spherical Bessel functions.

Considering the TE polarization of Bessel pincer light-sheet beam (beam order *l* and scaling parameter $\alpha_{0}$ with a time variation in the form of exp(−iωt)—suppressed from the subsequent equations for convenience) on a medium with a refractive index of $m_{0}$, a charged sphere of radius *a* and refractive index $m_{1}$ is placed in the field of the non-paraxial Bessel pincer light-sheet beam (similar with assimilation of [68]), as shown in Figure 1.

Through the multiple expansion of the incident plane wave and considering the orthogonal property of vector spherical wave functions (VSWFs [70]), the series expression of the incident plane wave is obtained as below:(2)$$E_{\mathrm{inc}}=-\sum_{n=1}^{\infty }\sum_{m=-n}^{n}iE_{mn}\left[p_{mn}N_{mn}^{\left(1\right)}\left(kr\right)+q_{mn}M_{mn}^{\left(1\right)}\left(kr\right)\right]$$
(3)$$H_{\mathrm{inc}}=-\frac{k}{\omega \mu }\sum_{n=1}^{\infty }\sum_{m=-n}^{n}E_{mn}\left[q_{mn}N_{mn}^{\left(1\right)}\left(kr\right)+p_{mn}M_{mn}^{\left(1\right)}\left(kr\right)\right]$$
where
(4)$$E_{mn}=i^{n}\left|E_{0}\right|{\left[\frac{2n+1}{n(n+1)}\frac{(n-m)!}{(n+m)!}\right]}^{1/2}$$
(5)$$M_{mn}^{\left(1\right)}\left(kr\right)=\left[i\pi_{mn}\left(\cos\theta \right)e_{\theta }-\tau_{mn}\left(\cos\theta \right)e_{\phi }\right]j_{n}\left(kr\right)e^{im\phi }$$
(6)$$\begin{array}{c} N_{mn}^{\left(1\right)}\left(kr\right)\;\;\;\;\;=\left[\tau_{mn}\left(\cos\theta \right)e_{\theta }+i\pi_{mn}\left(\cos\theta \right)e_{\phi }\right]\frac{1}{kr}\frac{d}{dr}\left[rj_{n}\left(kr\right)\right]e^{im\phi }\\ \;\;\;\;\;\;\;\;\;\;\;\;\;\;\;\;\;\;\;\;\;\;\;\;\;\;\;\;\;\;\;\;\;\;\;\;\;\;\;\;\;\;\;+e_{r}n\left(n+1\right)P_{n}^{m}\left(\cos\theta \right)\frac{j_{n}\left(kr\right)}{kr}e^{im\phi } \end{array}$$
with
(7)$$p_{mn}=k\frac{i^{1-m}}{E_{0}}D_{mn}^{1/2}\int_{\alpha =0}^{\pi /2}d\alpha e^{-ik\cos\alpha z_{0}}S_{x}\left(\alpha \right)\pi_{mn}\left(\cos\alpha \right)\cos\alpha$$
(8)$$q_{mn}=k\frac{i^{1-m}}{E_{0}}D_{mn}^{1/2}\int_{\alpha =0}^{\pi /2}d\alpha e^{-ik\cos\alpha z_{0}}S_{x}\left(\alpha \right)\tau_{mn}\left(\cos\alpha \right)\cos\alpha$$
(9)$$D_{mn}=\frac{\left(2n+1\right)\left(n-m\right)!}{n(n+1)(n+m)!}$$
(10)$$\pi_{mn}\left(\cos\theta \right)=\frac{m}{\sin\theta }P_{n}^{m}\left(\cos\theta \right)\;$$
(11)$$\tau_{mn}\left(\cos\theta \right)=\frac{dP_{n}^{m}\left(\cos\theta \right)}{d\theta }$$
where $P_{n}^{m}\left(\cos\alpha \right)$ represents the associated Legendre polynomials of degree *n* and order *m*; $j_{l}\left(\cdot \right)$ is the spherical Bessel functions of *l* order; and $e_{\left(r,\theta ,\phi \right)}$ are radial, polar, and azimuthal unit vectors.

According to GLMT [71], the internal filed $E_{\mathrm{int}}$ is also expanded in terms of the VSWFs ($N_{mn}^{\left(1\right)}\left(k,r\right)\mathrm{and}M_{mn}^{\left(1\right)}\left(k,r\right)$) and the scattered field $E_{s\mathrm{ca}}$ in terms of the VSWFs ($N_{mn}^{\left(3\right)}\left(k,r\right)$ and $M_{mn}^{\left(3\right)}\left(k,r\right)$).
(12)$$E_{\mathrm{int}}=-\sum_{n=1}^{\infty }\sum_{m=-n}^{n}iE_{mn}\left[d_{mn}N_{mn}^{\left(1\right)}\left(m_{1}kr\right)+c_{mn}M_{mn}^{\left(1\right)}\left(m_{1}kr\right)\right]$$
(13)$$H_{\mathrm{int}}=-\frac{k_{1}}{\omega \mu_{1}}\sum_{n=1}^{\infty }\sum_{m=-n}^{n}E_{mn}\left[c_{mn}N_{mn}^{\left(1\right)}\left(kr\right)+d_{mn}M_{mn}^{\left(1\right)}\left(kr\right)\right]$$
(14)$$E_{s\mathrm{ca}}=\sum_{n=1}^{\infty }\sum_{m=-n}^{n}iE_{mn}\left[a_{mn}N_{mn}^{\left(3\right)}\left(kr\right)+b_{mn}M_{mn}^{\left(3\right)}\left(kr\right)\right]$$
(15)$$H_{s}=\frac{k}{\omega \mu }\sum_{n=1}^{\infty }\sum_{m=-n}^{n}E_{mn}\left[b_{mn}N_{mn}^{\left(3\right)}\left(kr\right)+a_{mn}M_{mn}^{\left(3\right)}\left(kr\right)\right]$$
with
(16)$$M_{mn}^{\left(3\right)}=\left[i\pi_{mn}\left(\cos\theta \right)e_{\theta }-\tau_{mn}\left(\cos\theta \right)e_{\phi }\right]h_{n}^{\left(1\right)}\left(kr\right)e^{im\phi }$$
(17)$$\begin{array}{c} N_{mn}^{\left(3\right)}=\left[i\tau_{mn}\left(\cos\theta \right)e_{\theta }+\pi_{mn}\left(\cos\theta \right)e_{\phi }\right]\frac{1}{kr}\frac{d}{dr}\left[rh_{n}^{\left(1\right)}\left(kr\right)\right]e^{im\phi }\\ +e_{r}n\left(n+1\right)P_{n}^{m}\left(\cos\theta \right)\frac{h_{n}^{\left(1\right)}\left(kr\right)}{kr}e^{im\phi } \end{array}$$

The incident field, scattering field, and internal field at the boundary between the charged sphere [60,72] and the air meet the following conditions:(18)$$\begin{array}{cc} n\cdot (\varepsilon_{0}E_{2}-\varepsilon_{1}E_{1}) & =\eta \\ n\cdot (\mu_{0}H_{2}-\mu_{1}H_{1}) & =0\\ n\times (E_{2}-E_{1}) & =0\\ n\times (H_{2}-H_{1}) & =K\\ \sigma E_{1}\cdot n-\nabla \cdot K & =\frac{\partial \eta }{\partial t} \end{array}$$
where σ is the volume conductivity of the sphere; *n* is the unit vector perpendicular from medium 1 (interior space of the charged spherical particle) to medium 2 (outer space of the charged spherical particle); and $E_{1}$, $E_{2}$ and $H_{1}$, $H_{2}$ are the total electric field and the total magnetic field, respectively. $\mu_{0}$ and $\mu_{1}$ are the magnetic permeability in vacuum and medium, respectively. $\epsilon_{0}$ and $\epsilon_{1}$ are the permittivity in vacuum and medium, respectively. η is surface current density, K is surface current, with the relation $K=\sigma_{s}E_{l,\tau }$. $E_{l,\tau }$ is the tangential component of the electric field on the surface.

The surface conductivity $\sigma_{s}$ of the sphere is expressed as
(19)$$\sigma_{s}\left(\omega \right)=\frac{i\eta e/m_{e}}{\omega +i\gamma_{s}}$$
where *e* = 1.629 × $10^{-19}C$ is the electron charge and $m_{e}$ = 9.109 × $10^{-31}$ kg is the electron mass. $\gamma_{s}\approx k_{B}T/\hbar$ is the classical parameter (derived by J.Kl$\overset{⌣}{\;a}$cka and M. Kocifaj), *T* is the sphere temperature, $k_{B}$ is the Boltzmann’s constant, and $\hbar =1.0546\times 10^{-34}$ Js is the Plank constant divided by 2π.

Substituting the electromagnetic field (Equations (Equation 2), (Equation 3), (Equation 12)–( Equation 15)) into the boundary conditions, Equation (Equation 18), the partial wave expansion coefficients of the scattering field $a_{mn}$ and $b_{mn}$, and the internal field $c_{mn}$ and $d_{mn}$, can be obtained
(20)$$a_{mn}=a_{n}p_{mn},\;\;b_{mn}=b_{n}q_{mn}$$
(21)$$c_{mn}=c_{n}q_{mn},\;\;d_{mn}=d_{n}p_{mn}$$
where $a_{n}$ and $b_{n}$ are the Mie scattering coefficients of charged sphere [60,72,73,74,75], and $c_{n}$ and $d_{n}$ are the corresponding internal coefficients.

And,
(22)$$a_{n}=\frac{\psi_{n}\left(x\right)\psi_{n}^{\prime }\left(mx\right)-m\psi_{n}\left(mx\right)\psi_{n}^{\prime }\left(x\right)-i\omega \mu_{0}\sigma_{s}\psi_{n}^{\prime }\left(mx\right)\psi_{n}^{\prime }\left(x\right)/k}{\xi_{n}\left(x\right)\psi_{n}^{\prime }\left(mx\right)-m\psi_{n}\left(mx\right)\xi_{n}^{\prime }\left(x\right)-i\omega \mu_{0}\sigma_{s}\psi_{n}^{\prime }\left(mx\right)\xi_{n}^{\prime }\left(x\right)/k}$$
(23)$$b_{n}=\frac{\psi_{n}\left(mx\right)\psi_{n}^{\prime }\left(x\right)-m\psi_{n}\left(x\right)\psi_{n}^{\prime }\left(mx\right)+i\omega \mu_{0}\sigma_{s}\psi_{n}\left(x\right)\psi_{n}\left(mx\right)/k}{\psi_{n}\left(mx\right)\xi_{n}^{\prime }\left(x\right)-m\xi_{n}\left(x\right)\psi_{n}^{\prime }\left(mx\right)+i\omega \mu_{0}\sigma_{s}\psi_{n}\left(mx\right)\xi_{n}\left(x\right)/k}$$
(24)$$c_{n}=\frac{m\psi_{n}\left(x\right)\xi_{n}^{\prime }\left(x\right)-m\xi_{n}\left(x\right)\psi_{n}^{\prime }\left(x\right)}{\psi_{n}\left(mx\right)\xi_{n}^{\prime }\left(x\right)-m\xi_{n}\left(x\right)\psi_{n}^{\prime }\left(mx\right)+i\omega \mu_{0}\sigma_{s}\psi_{n}\left(mx\right)\xi_{n}\left(x\right)/k}$$
(25)$$d_{n}=\frac{m\xi_{n}\left(x\right)\psi_{n}^{\prime }\left(x\right)-m\psi_{n}\left(x\right)\xi_{n}^{\prime }\left(x\right)}{\xi_{n}\left(x\right)\psi_{n}^{\prime }\left(mx\right)-m\psi_{n}\left(mx\right)\xi_{n}^{\prime }\left(x\right)-i\omega \mu_{0}\sigma_{s}\psi_{n}^{\prime }\left(mx\right)\xi_{n}^{\prime }\left(x\right)/k}$$

Here, $\psi_{n}\left(x\right)=xj_{n}\left(x\right)$ and $\xi_{n}\left(x\right)=xh_{n}^{1}\left(x\right)$, in which $j_{n}\left(x\right)$ and $h_{n}^{1}\left(x\right)$ are the first-order Bessel function and the spherical Hankel function, respectively. x=ka and $m=m_{1}/m_{0}$ are the size parameter and the refractive index, respectively. $\mu_{0}$ is the permeability in vacuum. The prime is the derivative of the functions.

The scattering electric field [76,77] $E_{s\mathrm{ca}}$ can be expressed by the scattering amplitude functions, $S_{1}$ and $S_{2}$,
(26)$$\left.\left(\begin{array}{c} E_{s\theta }\\ E_{s\varphi } \end{array}\right.\right)=\frac{e^{ik(r-z)}}{-ikr}\left(\begin{array}{cc} S_{2} & 0\\ 0 & S_{1} \end{array}\right)\left(\begin{array}{c} E_{i\theta }\\ E_{i\varphi } \end{array}\right)$$
with
(27)$$S_{1}=\sum_{n=1}^{\infty }\sum_{m=-n}^{n}E_{mn}{\left(-i\right)}^{n}\left[a_{mn}\pi_{mn}\left(cos\theta \right)+b_{mn}\tau_{mn}\left(cos\theta \right)\right]\exp\left(im\phi \right)$$
(28)$$S_{2}=\sum_{n=1}^{\infty }\sum_{m=-n}^{n}E_{mn}{\left(-i\right)}^{n}\left[a_{mn}\tau_{mn}\left(\cos\theta \right)+b_{mn}\pi_{mn}\left(\cos\theta \right)\right]\exp\left(im\phi \right)$$

Scattering efficiency refers to the degree to which incident light is scattered by a material; extinction efficiency refers to the effect of incident light being fully or partially extinguished after passing through a substance; and absorption efficiency refers to the degree to which a material absorbs incident light. Once the scattering coefficients and BSCs are determined, the scattering $Q_{sca}$, extinction $Q_{ext}$, and absorption $Q_{abs}$ are obtained by dividing the cross-section.
(29)Qsca=Cscaπa2=4k2a2∑n=1∞∑m=−nnamn2+bmn2Qext=Cextπa2=4k2a2Re∑n=1∞∑m=−nnqmnbmn*+pmnamn*Qabs=Qext−Qsca
where $C_{abs}$ and $C_{ext}$ [64,68] are the scattering and extinction cross sections, respectively. The star * denotes the complex conjugate.

## 3. Results

The calculations are made for spherical particle (surrounded by air with a refractive index $m_{2}$ = 1.0) assuming that the complex refractive index is $m_{1}$ = $1.33+10^{-6}i$, with temperature *T* = 300 K. The incident beam wavelength is λ=0.6328 μm. To obtain large enough ka to ensure adequate convergence of Equations (Equation 27)–(Equation 29), $n_{\max}=ka+4{\left(ka\right)}^{1/3}+10$ is generally used. The electric field intensity is normalized and set as $E_{0}$ = 1.

### 3.1. Far-Field Scattering Intensity

The intensity of scattered light at a certain angle is often used to retrieve direct or other information about particles in the optical measurement of particles. Considering the size parameter ka, beam wavelength λ, azimuth angle ϕ, surface charge $\sigma_{s}$, refractive index $m_{1}$, beam order ł, and beam scaling parameter $\alpha_{0}$, the scattering intensity expressed as $I\left(r,\theta ,\phi \right)=\frac{1}{k^{2}r^{2}}\left[{\left|S_{1}\right|}^{2}+{\left|S_{2}\right|}^{2}\right]$ is employed to calculate the far-field scattered intensity for the Bessel pincer light-sheet beam on the sphere (charged one and uncharged one). Note that the size parameter is x=ka.

#### 3.1.1. Intensity I with Different Size Parameter ka

Figure 2 shows that the far scattering varies with different size parameters ka, and the sphere particle varies from smaller to larger. Figure 2a is the polar diagram while Figure 2b is the Cartesian graphic. In polar coordinates, the scattering intensity I at the forward scattering angle ($0^{\circ }$–$180^{\circ }$) increases as the size parameter *x* increases. Additionally, the difference between the charged and uncharged sphere decreases with increasing *x*. The impact of the size parameter *x* on the scattering intensity in all directions is more clearly displayed and visualized in the Cartesian coordinate system, providing a more intuitive representation. For the curve of x=0.6, the scattering intensity amplitude of the charged sphere is higher than that of the neutral sphere and the resonance peak of the charged sphere is enhanced relative to the neutral sphere. For the curve of x=5, the scattering intensity amplitude of the charged sphere is slightly higher than that of the neutral sphere and the resonance small peak distribution of the charged sphere is enhanced relative to the neutral sphere. For the curve of x=25, the scattering intensity amplitude of the charged sphere is comparable to that of the neutral sphere and the resonance small peak distribution of the charged sphere is almost the same as that of the neutral sphere. The scattering intensity amplitude increases as *x* increases, and the number of resonance peaks increases with *x*.

#### 3.1.2. Intensity I with Different Beam Wavelength λ

Figure 3 demonstrates how the scattering intensity changes with differing scattering angles, showcasing the impact of varying wavelengths (f = [0.24, 2.0, 24.0] THz) and different particle size parameters (*x* = [0.6, 5, 25]). Figure 3a,b depict the scattering intensity distribution patterns for different incident wave frequencies, highlighting the particle size parameter set at x=0.6 in both polar and Cartesian coordinate systems. At f = 0.24 THz, the scattering intensity amplitude of the charged sphere varies significantly from that of the neutral sphere, yet there is no noticeable difference in the resonance peak position. At f = 2.0 THz, the scattering intensity amplitude increases dramatically by four orders of magnitude compared to f = 0.24 THz without a shift in the resonance peak position. Similarly, for f = 24 THz, the scattering intensity amplitude of the charged neutral sphere increases dramatically by four orders of magnitude compared to f = 2.0 THz, with the resonant peak position remaining unchanged. Figure 3c,d illustrate the scattering intensity distribution patterns for varying incident wave frequencies, emphasizing the particle size parameter set at x=5 in both polar and Cartesian coordinate systems. When *x* = 5, the amplitude of the scattering intensity for charged and neutral particles under the exertion of the Bessel pincer beam increases significantly compared to that of x=0.6. The resonance peaks are generated at approximately the same scattering angle. However, unlike the scenario at *x* = 0.6, the difference in scattering intensity between charged particles and neutral particles is significantly reduced at f = 24 THz. Figure 3e,f demonstrate the scattering intensity distribution patterns across different incident wave frequencies, with a particular emphasis on the particle size parameter being *x* = 0.6 in both polar and Cartesian coordinate systems. Similarly, with x=25, there is a substantial increase in the scattering intensity amplitude while the position of the resonance peak remains unchanged. However, for the curve at f = 24 THz, there is almost no disparity in the trend of scattering intensity between charged spheres and neutral spheres. With an increase in the parameter *x* across different scales, the scattering intensity amplitudes of both charged and neutral spheres also rise. Likewise, as the frequency of the incident wave escalates, the scattering intensity amplitude experiences a significant increase. Nevertheless, under high-frequency incident waves and large-size parameters *x*, the trend of scattering intensity on charged and neutral particles remains largely consistent.

#### 3.1.3. Intensity I with Different Azimuth Angle ϕ

Figure 4 delineates the progression of far-field scattering intensity concerning changes in scattering angles, considering diverse azimuth angle ϕ adjustments for varying size parameters *x*. For the particle size parameter x=0.6 (as shown in Figure 4a,b), the scattering intensity varies across different azimuth angles. Specifically, when the azimuth angle is less than π/2, the scattering intensity shows a certain degree of dependence on the azimuth angle, gradually strengthening as the azimuth angle decreases. When the particle size parameter is x=5 (as shown in Figure 4c,d), there is a more pronounced dependence of the scattering intensity distribution on the azimuth angle compared to x=0.5. Furthermore, at x=25 (as demonstrated in Figure 4e,f), the azimuth angle exhibits a heightened influence on the scattering intensity distribution, surpassing the dependency observed at x=5, and the scattering intensity amplitude also increases by nearly an order of magnitude.

#### 3.1.4. Intensity I with Different Surface Charge $\sigma_{s}$

Considering the Bessel pincer light-sheet beam on the charge particle with different surface charge $\sigma_{s}$, Figure 5 shows the effect of the particle surface charge on the scattering intensity as the scattering angle changes in the polar coordinate system and the Cartesian coordinate system. When the particle size parameter is x=0.6 (as shown in Figure 5a,b), for the particle surface charge $\sigma_{s}<4\times 10^{-8}$, its far-field scattering intensity is almost the same as the distribution of neutral particles; when the charged particle is $\sigma_{s}=5\times 10^{-4}$, the far-field scattering intensity amplitude is greater than that of neutral particles and small surface charges, and the number of corresponding prominent resonance peaks is mainly two, which is precisely the same as the number of prominent resonance peaks on neutral particles, but the position of the resonance peak is shifted. For the particle size parameter x=5 (as shown in Figure 5c,d), when the surface charge is less than $\sigma_{s}<4\times 10^{-8}$, the far-field scattering intensity maintains a consistent distribution and more resonance peaks appear. Especially, while the scattering angle is between approximately 130° and 230°, for the curve $\sigma_{s}=5\times 10^{-4}$, the resonance intensity is obviously much higher than the curve of $\sigma_{s}=0$. However, for the particle size parameter x=25 (as shown in Figure 5e,f), compared to the cases where the particle size is x=5 and x=0.6, the far-field scattering intensity distribution trends of charged particles and neutral particles are consistent, with more minor differences.

#### 3.1.5. Intensity I with Different Refractive Index $m_{1}$

Considering the imaginary part $n_{p}$ of the refractive index of spherical particles ($m_{1}$), Figure 6 illustrates the far-field scattering intensity distribution of charged and neutral particles at various scattering angles when subjected to Bessel pincer light-sheet beams with different scale parameters. The far-field scattering intensity distribution trend is minimally affected by the change in the imaginary part when x=0.6 (as shown in Figure 6a,b) and the real part of the refractive index is not maintained. Specifically, for $n_{p}$ < 0.5, charging has little impact on far-field scattering intensity; for $n_{p}$ = 0.5, charged spheres exhibit slightly higher far-field scattering intensity than neutral particles. Moreover, at x=5 (as shown in Figure 6c,d), both charged and neutral particles show a significant increase in far-field scattering intensity amplitude and resonance peak. Notably, when $n_{p}$ = 0, particle scattering intensity (charged or not) surpasses that of absorbing particles. Similarly, at x=25 (as shown in Figure 6e,f), the scattering amplitude resembles that of x=5 but with an overall increase in resonance peak for far-field scattering intensity. Unlike the case of x=5, there is minimal difference in the scattering field distribution trend between 90° and 270°.

#### 3.1.6. Intensity I with Different Beam Order ł

The variation trend of far-field scattering intensity with scattering angle is depicted in Figure 7 for Bessel pincer light-sheet beams with different size parameters and beam order ł, acting on a charged sphere particle in both polar and Cartesian coordinates. When the particle size parameter x=0.6 (as shown in Figure 7a,b) and the beam order ł remains unchanged, the far-field scattering intensity distribution of charged particles and neutral particles exhibits a similar pattern. Furthermore, an increase in beam order ł results in a consistent trend of increasing far-field scattering intensity distribution. Similarly, when the particle scale parameter is x=5 ((as shown in Figure 7c,d)), the increase in beam order is directly proportional to the far-field scattering intensity. However, a difference exists between the far-field scattering intensities of charged and neutral particles under the same beam order. In contrast to x=5, when the particle size parameter is x=25 (as shown in Figure 7e,f), there is a significant increase in resonant peaks concerning the scattering angle for both charged spheres and neutral spheres.

#### 3.1.7. Intensity I with Different Beam Scaling Parameter $\alpha_{0}$

The variation trend of far-field scattering intensity with scattering angle is depicted in Figure 8 for Bessel pincer light-sheet beams with different size parameters and beam scaling parameter $\alpha_{0}$ values, acting on a charged sphere particle in both polar and Cartesian coordinates. When the particle size parameter is x=0.6 (as shown in Figure 8a,b), the scattering intensity distribution trend of the charged sphere corresponds to that of the neutral sphere with the same beam scaling parameter $\alpha_{0}$ curve. Still, the wave peak corresponding to the charged sphere scattering is higher. Furthermore, as $\alpha_{0}$ increases, charged and neutral spheres gradually increase in the far-field scattering intensity wave crest. In contrast, for particle size parameters x=5 (as shown in Figure 8c,d) and x=25 (as shown in Figure 8e,f), their far-field scattering intensity distributions are essentially identical to that of x=0.6; however, there is an increasing number of resonant peaks with larger particle sizes.

### 3.2. Efficiency Factor

Considering the influence of refractive index $m_{1}$, beam order ł, beam scaling parameter $\alpha_{0}$, and surface charge $\sigma_{s}$, the scattering, absorption, and extinction efficiencies for a charged particle illuminated by a Bessel pincer light-sheet beam are analyzed. The wavelength of the incident beam is λ=0.6328 μm. The temperature is 300 K. The dimensionless size parameter of the sphere ranges from 0.1 to 20.

#### 3.2.1. Efficiencies with Different Refractive Index $m_{1}$ under Varying Size Parameter ka

The changes in scattering, absorption, and extinction efficiency of charged particles are depicted in Figure 9, as the imaginary refractive index $n_{p}$ varies under a Bessel pincer light-sheet’s beam illumination with increasing dimensionless size parameters. In Figure 9a, when particle absorption is zero (i.e., $n_{p}$ = 0), the $Q_{ext}$ of charged particles and neutral particles is similar. As $n_{p}$ increases, the resonant peak of $Q_{ext}$ for charged and neutral particles gradually shifts towards the smaller dimensionless size parameter *x*. Additionally, when $n_{p}$ = 0.5, both charged and neutral particles exhibit higher extinction efficiency and resonate earlier. In contrast to the scattering efficiency shown in Figure 9b, resonance and $Q_{sca}$ for both charged and neutral particles are highest when absorption is zero. In Figure 9c, for $n_{p}$ = 0, resonance for charged particles intensifies in regions with small dimensionless size parameter *x*; as absorption increases (i.e., an increase in $n_{p}$), resonance is primarily enhanced at the central peak.

#### 3.2.2. Efficiencies with Different Beam Scaling Parameter $\alpha_{0}$ under Varying Size Parameter ka

Figure 10 shows the scattering, absorption, and extinction efficiency of charged particles illuminated by a Bessel pincer light-sheet beam of different beam scaling parameter values $\alpha_{0}$, varying with dimensionless size parameter ka. When the beam scaling parameters $\alpha_{0}$ are held constant, the difference in $Q_{ext}$ (as shown in Figure 10a), $Q_{sca}$ (as shown in Figure 10b), and $Q_{abs}$ (as shown in Figure 10c) between charged particles and neutral particles is small. With the increase in beam scaling parameters $\alpha_{0}$, the $Q_{ext}$, $Q_{sca}$, and $Q_{abs}$ of charged particles and neutral particles show a changing trend. First, they reach a resonance peak, then gradually converge with the increase in dimensionless size parameters. Unlike the neutral sphere, the $Q_{abs}$ (as shown in Figure 10c) of the charged sphere first reaches a small resonance peak before reaching the resonance peak.

#### 3.2.3. Efficiencies with Different Beam Order ł under Varying Size Parameter ka

Similar to Figure 10, the different beam order ł is considered in Figure 11. When increasing beam order ł, $Q_{ext}$, $Q_{sca}$, and $Q_{abs}$ on charged and neutral particles gradually enhanced with dimensionless parameter ka. For the cases of dimensionless size parameter x<10, $Q_{ext},\;Q_{sca}$, and $Q_{abs}$ of charged particle and neutral particle is almost always equal to 0. For the cases where x>10, there is virtually no difference in the $Q_{ext}$ (shown in Figure 11a) between charged particles and neutral particles. but there is a tiny difference in $Q_{sca}$ (shown in Figure 11b) and $Q_{abs}$ (shown in Figure 11c) between charged particle and neutral particle. The $Q_{sca}$ of the neutral particle is higher than that of the charged one for a different beam order, while the $Q_{abs}$ of the neutral particle is lower than that of the charged one for a different beam order.

#### 3.2.4. Efficiencies with Different Refractive Index $m_{1}$ under Varying Sigma $\sigma_{s}$

Figure 12 illustrates the variation in $Q_{ext}$ (shown in Figure 12a), $Q_{sca}$ (shown in Figure 12b), and $Q_{abs}$ (shown in Figure 12c) for Bessel pincer light-sheet beam on a charged particle sphere (with different imaginary part $n_{p}$ of refractive index) as the charge carried increases. When the surface charge $\sigma_{s}$ < 0.5, $Q_{ext}$ and $Q_{abs}$ of the charged particle increase with the increase in imaginary parts $n_{p}$ of different refractive indices. Further, with the rise in $\sigma_{s}$, there is a negative correlation trend. On the contrary, $Q_{sca}$ of the charged particle presents a positive correlation trend with the surface charge $\sigma_{s}$ increase. Furthermore, when the surface charge $\sigma_{s}$ > 0.5, $Q_{ext}$ and $Q_{abs}$ of the charged particle maintain the same value for different imaginary parts of the refractive index and present a ‘log’ curve growth with the increase in $\sigma_{s}$.

#### 3.2.5. Efficiencies with Different Beam Scaling Parameter $\alpha_{0}$ under Varying Sigma $\sigma_{s}$

Figure 13 illustrates the variation in $Q_{ext}$ (shown in Figure 13a), $Q_{sca}$ (shown in Figure 13b), and $Q_{abs}$ (shown in Figure 13c) for a Bessel pincer light-sheet beam with different beam parameters $\alpha_{0}$ on a charged particle sphere as the charge carried increases. With the beam scaling parameter $\alpha_{0}$ increase, both $Q_{ext}$ and $Q_{abs}$ gradually increase with the rise in surface conductivity initially and progressively converge to zero after reaching their peak values. Meanwhile, $Q_{sca}$ increases with the increase in surface conductivity and tends to plateau after reaching its peak value.

#### 3.2.6. Efficiencies with Different Beam Order ł under Varying Sigma $\sigma_{s}$

Figure 14 depicts the evolution of $Q_{ext}$ (shown in Figure 14a), $Q_{sca}$ (shown in Figure 14b), and $Q_{abs}$(shown in Figure 14c) for a Bessel pincer light-sheet beam with different beam order ł on a charged particle sphere as the surface charge is incremented. As the beam order λ rises, for $\sigma_{s}<$ 1 $\times \;10^{-4}$, both $Q_{ext}$ and $Q_{abs}$ surge along with the surface conductivity augmentation. When 1 $\times \;10^{-4}$ < 0 $\sigma_{s}<$1 $\times \;10^{-2}$, $Q_{ext}$ and $Q_{abs}$ initially experience gradual growth before swiftly declining to zero after peaking. Conversely, $Q_{sca}$ increases with higher surface conductivity and stabilizes after reaching its peak value.

#### 3.2.7. Efficiencies with Different Size Parameter *X* under Varying Sigma $\sigma_{s}$

Figure 15 shows $Q_{ext}$ (as shown in Figure 15a), $Q_{sca}$ (as shown in Figure 15b), and $Q_{abs}$ (as shown in Figure 15c) of the charged particle with different spherical radii (*a* is set as from 0.1 $\times \;10^{-7}$ to 1.0 $\times \;10^{-7}$), varying with different surface charge $\sigma_{s}$ values ($\sigma_{s}$ is set as from 1 $\times \;10^{-10}$ to 10); Figure 15d–f are the three-dimensional renderings of Figure 15a–c, respectively. To facilitate the observation of the distribution of charged particles under different particle radii, the *y*-axis and *z*-axis of Figure 15d–f are normalized, in which $Q_{0}=$ 1 $\times \;10^{-7}$ and radius = a/$Q_{0}$. When $\sigma_{s}$ < 1 $\times \;10^{-5}$, the $Q_{ext}$, $Q_{sca}$, and $Q_{abs}$ values for spherical particles of the same radius remain unchanged as $\sigma_{s}$ increases. As the particle radius increases, the $Q_{ext}$, $Q_{sca}$, and $Q_{abs}$ values on spherical particles gradually rise. There is significant variation in $Q_{ext}$ for spherical particles with different radii, whereas the difference is smaller for $Q_{sca}$ compared to $Q_{ext}$. Additionally, when $\sigma_{s}$ > 1 $\times \;10^{-5}$, the $Q_{ext}$ of spherical particles increases with the rise in $\sigma_{s}$, with $Q_{ext}$ for different particle radii rapidly converging to a fixed value after reaching a peak. The $Q_{sca}$ of spherical particles increases gradually with increasing $\sigma_{s}$, and the $Q_{sca}$ for different particle radii converges to a fixed value. A distinction is observed in the behavior of $Q_{abs}$ for spherical particles, as it grows gradually with increasing $\sigma_{s}$. In contrast, the $Q_{abs}$ for various particle radii rapidly converges to zero after peaking.

## 4. Conclusions

In the framework of GLMT, the scattering influence of Bessel pincer light-sheet beams with a charged sphere is investigated. Utilizing the classical Mie scattering coefficient for a charged sphere and referencing the beam factor $p_{mn},q_{mn}$ of a Bessel pincer light-sheet beam from existing literature, the incident, scattering, and internal fields were expanded using Vector Spherical Wave Functions (VSWFs). Subsequently, calculations were made for far-field scattering intensity, extinction coefficient, scattering coefficient, and absorption coefficient. Results show significant differences in far-field scattering between charged and neutral spheres for small dimensionless parameters. Various factors such as order, dimension, wavelength, incident angle, and refractive index influence the scattering behavior of charged spheres. However, as dimensionless parameters increase, the differences between charged and neutral spheres diminish. The impact of refractive index and beam parameters on scattering, extinction, and absorption coefficients varies but tends to converge as dimensionless parameters increase. When Bessel pincer light-sheet beams with different parameters are applied to charged spheres with varying refractive indices, the efficiency in scattering, extinction, and absorption changes with surface charge variations. Once surface charge saturates, efficiencies stabilize. These findings are valuable for applications in particle manipulation (particle rotating, sorting devices, optical sectioning) and classification (particle characterization and sizing).

## Figures and Tables

**Figure 1 micromachines-15-00975-f001:**
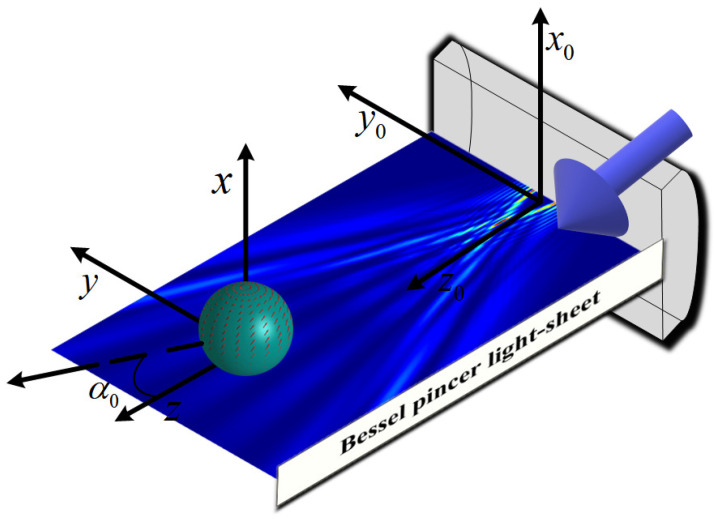
Distribution of the propagation of a non-paraxial Bessel pincer light-sheet on a charged sphere along *z*-direction (λ = 0.6328 μm, the radius *a* = 10 μm, ł = 10, $\alpha_{0}=0.5$).

**Figure 2 micromachines-15-00975-f002:**
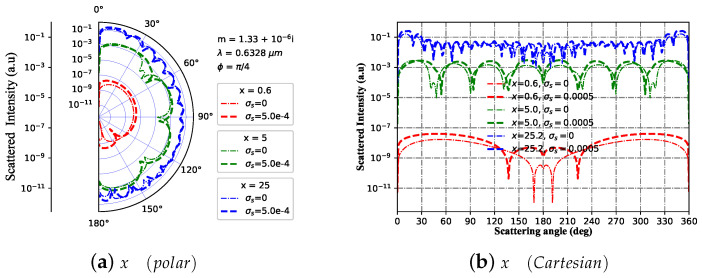
Scattered light intensity distribution varies with the scattering angle for a given particle size ka ([0.6, 5, 25]), in which $m_{1}$ = $1.33+10^{-6}i$ and $\sigma_{s}=5\times 10^{-4}$ S. The wavelength of incident beam is λ=0.6328 μm, beam scaling parameter is $\alpha_{0}=0.1$, beam order is ł=3, and the azimuth angle is ϕ=π/4.

**Figure 3 micromachines-15-00975-f003:**
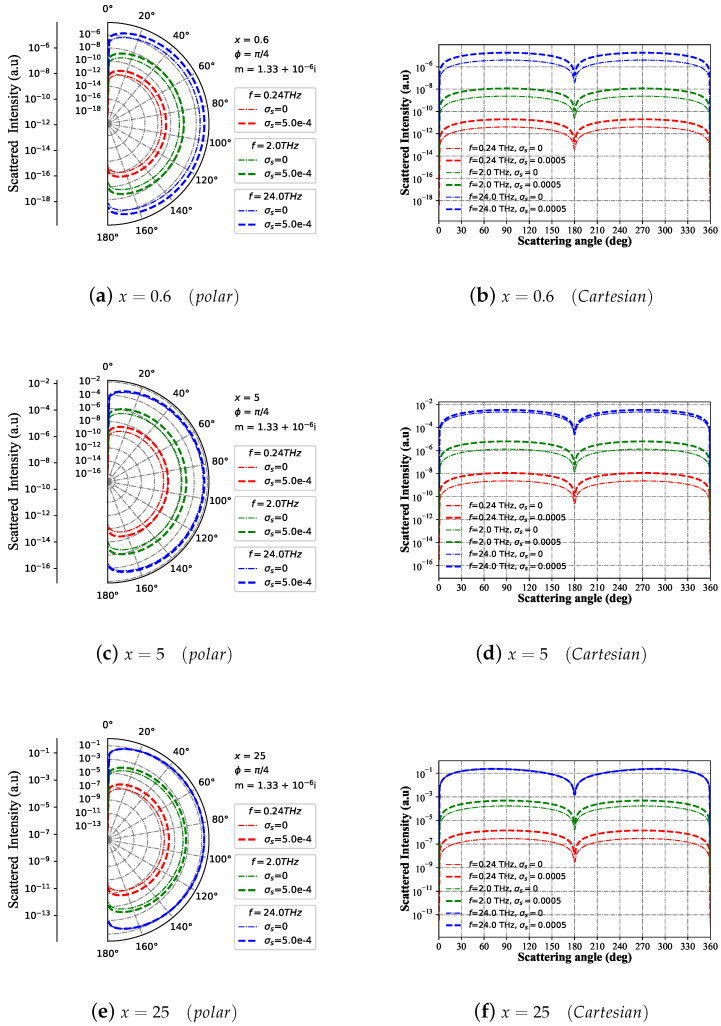
Far-field scattering intensity varies with the scattering angle for given charged particle ($m_{1}$ = $1.33+10^{-6}i$ and $\sigma_{s}=5\times 10^{-4}$ S) size parameters of 25, 5, and 0.6 under different beam wavelengths ($\lambda =v_{c}/f$; $v_{c}$ is the speed of light in vacuum). The frequency of incident beam is f=[0.24,2.0,24] THz, beam scaling parameter is $\alpha_{0}=0.1$, beam order is ł=3, and the azimuth angle is ϕ=π/4.

**Figure 4 micromachines-15-00975-f004:**
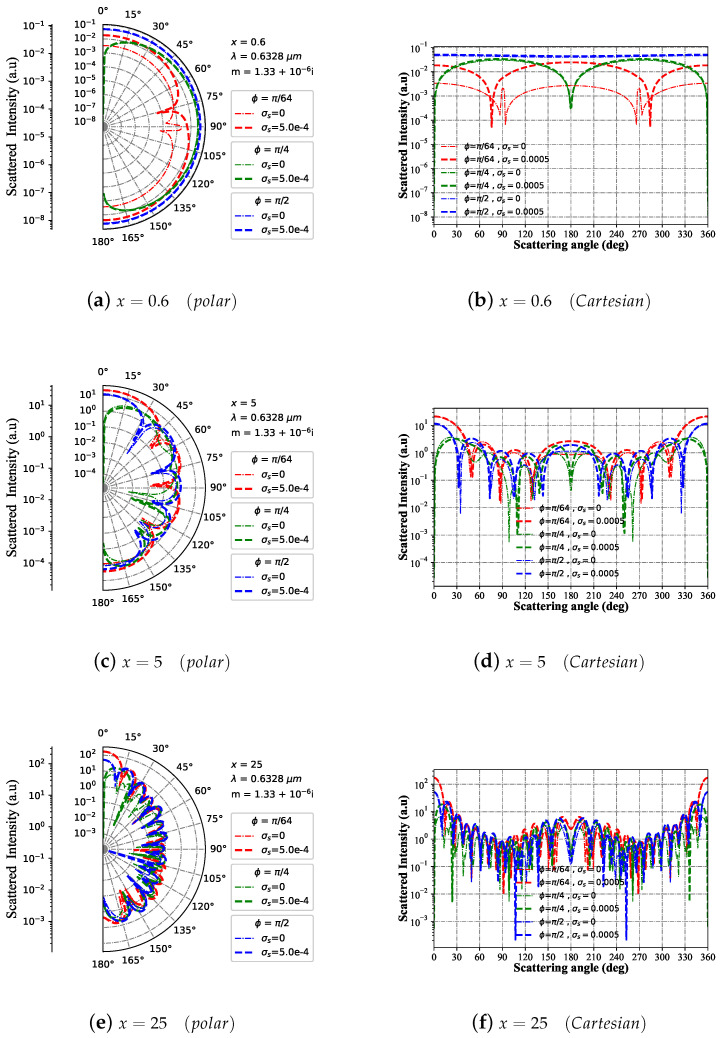
Far-field scattering intensity varies with the scattering angle for given charged particle ($m_{1}$ = $1.33+10^{-6}i$ and $\sigma_{s}=5\times 10^{-4}$ S) size parameters of 25, 5, and 0.6 under different azimuth angles (ϕ=[π/64,π/4,π/2]), respectively. The incident wavelength is λ=0.6328 μm, beam scaling parameter is $\alpha_{0}=0.1$, and beam order is ł=0.

**Figure 5 micromachines-15-00975-f005:**
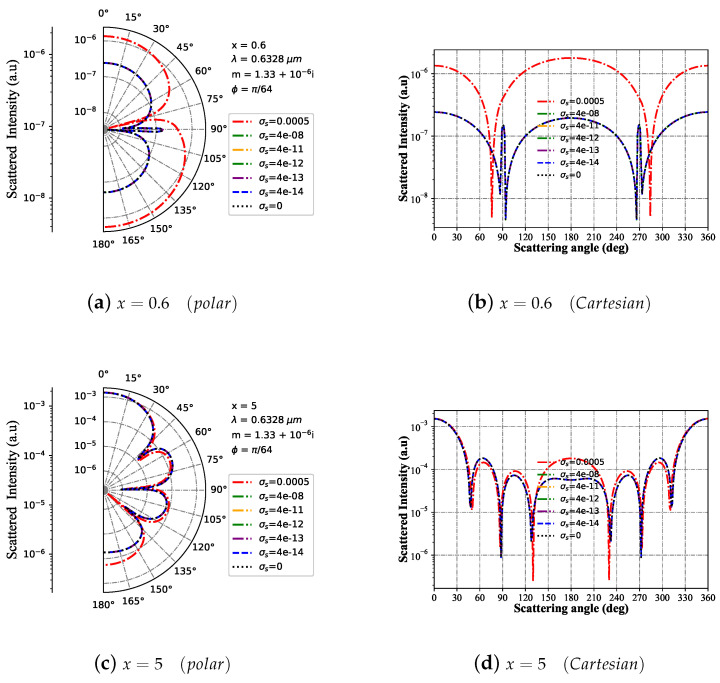
Far-field scattering intensity varies with the scattering angle for given charged particle ($m_{1}$ = $1.33+10^{-6}i$) size parameters of 25, 5, and 0.6 under different surface charges ($4\times 10^{-12}$, $\sigma_{s}$ = [$5\times 10^{-4}$, $4\times 10^{-8}$, $4\times 10^{-11}$, $4\times 10^{-12}$, $4\times 10^{-13}$, $4\times 10^{-14}$, 0]), respectively. The incident wavelength is λ=0.6328 μm, beam scaling parameter is $\alpha_{0}=0.1$, beam order is ł=10, and the azimuth angle is ϕ=π/64.

**Figure 6 micromachines-15-00975-f006:**
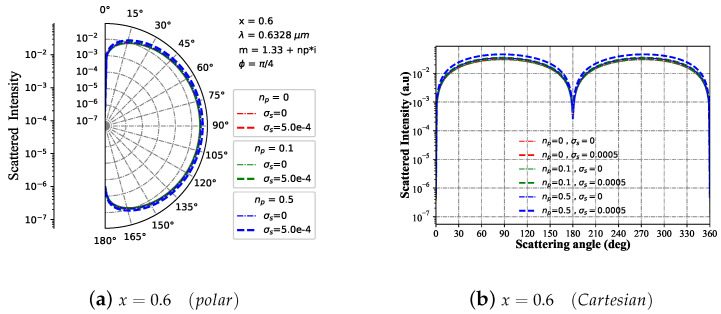
Far-field scattering intensity varies with the scattering angle for given charged particle size parameters of 25, 5, and 0.6 under different refractive indexes ($m_{1}=1.33+np*i$, with $\;n_{p}=\left[0,\;0.1,\;0.5\right]$), respectively. The incident wavelength is λ=0.6328 μm, beam scaling parameter is $\alpha_{0}=1.0$, beam order is ł=0, and the azimuth angle is ϕ=π/4.

**Figure 7 micromachines-15-00975-f007:**
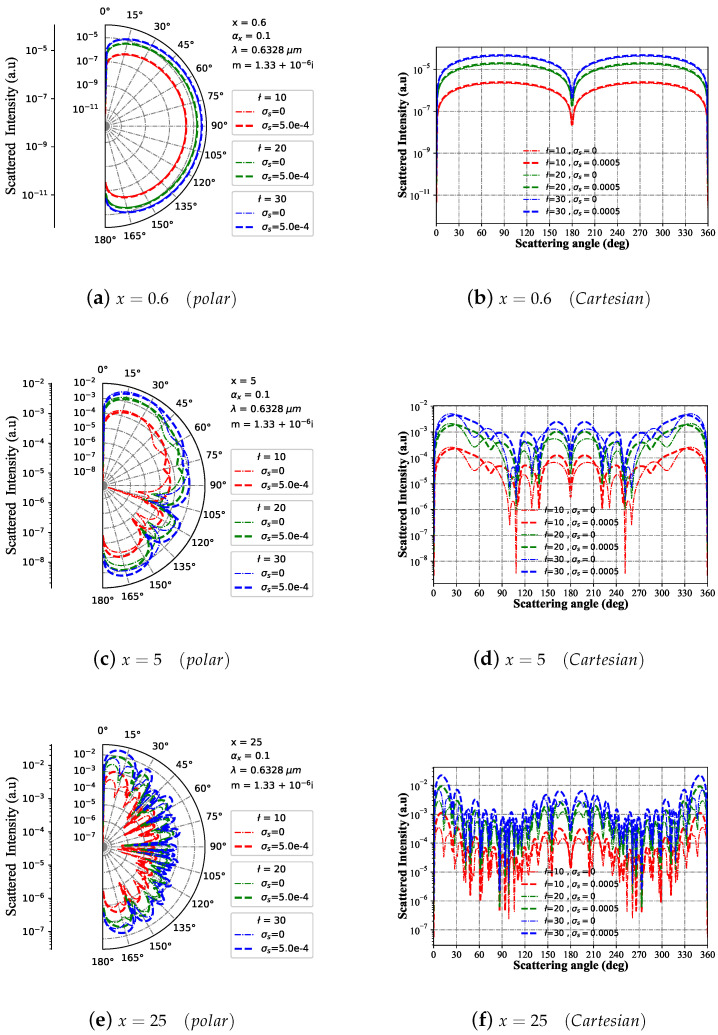
Far-field scattering intensity varies with the scattering angle for given charged particle ($m_{1}$ = $1.33+10^{-6}i$ and $\sigma_{s}=5.0\times 10^{-4}\;S$) size parameters of 25, 5, and 0.6 under different beam orders (ł=[10,20,30]), respectively. The incident wavelength is λ=0.6328 μm, beam scaling parameter is $\alpha_{0}=0.1$, and the azimuth angle is ϕ=π/4.

**Figure 8 micromachines-15-00975-f008:**
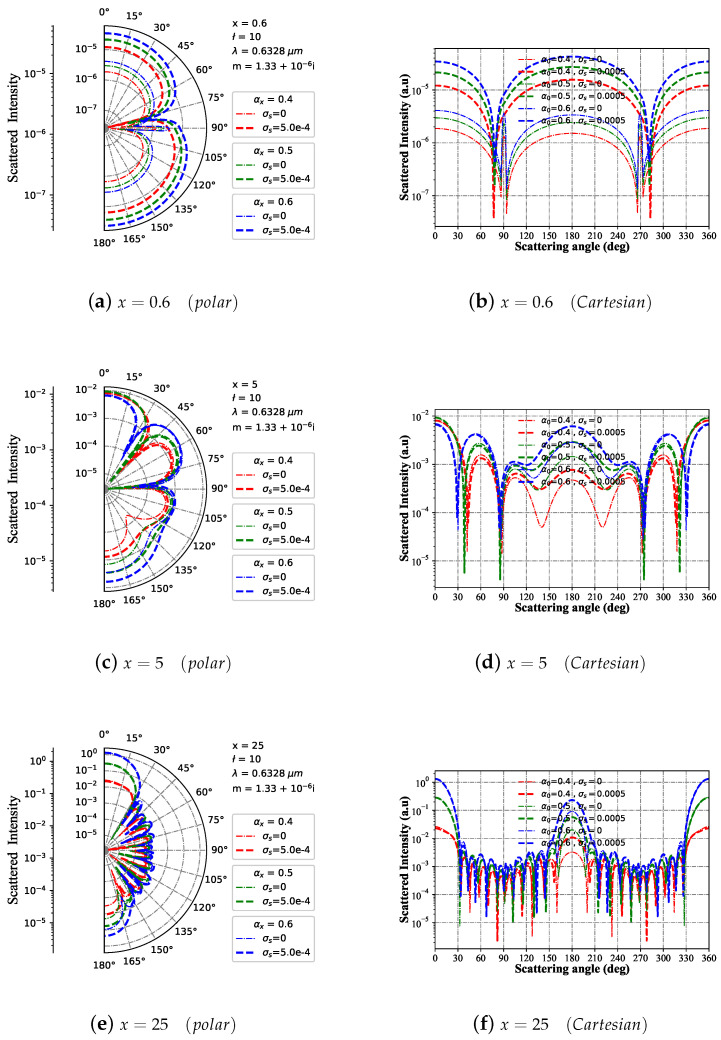
Far-field scattering intensity varies with the scattering angle for given charged particle ($m_{1}$ = $1.33+10^{-6}i$ and $\sigma_{s}=5.0\times 10^{-4}\;S$) size parameters of 25, 5, and 0.6 under different beam scaling parameter values ($\alpha_{0}=\left[0.4,\;0.5,\;0.6\right]$), respectively. The incident wavelength is λ=0.6328 μm, beam order is ł=10, and the azimuth angle is ϕ=π/4.

**Figure 9 micromachines-15-00975-f009:**
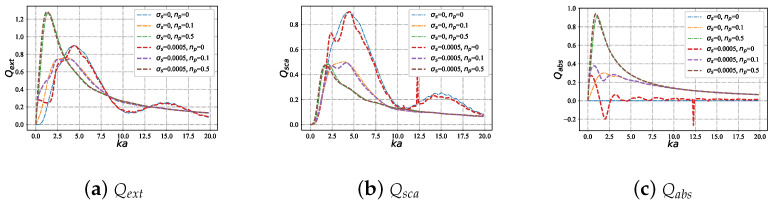
$Q_{ext},Q_{sca},Q_{abs}$ on a charged particle illuminated by a Bessel pincer light-sheet beam (beam order ł=0, beam scaling parameter $\alpha_{0}=1.0$, and wavelength λ=0.6328 μm) under different refractive indexes $n_{p}$ (in which $m_{1}=1.33+j*n_{p}$).

**Figure 10 micromachines-15-00975-f010:**
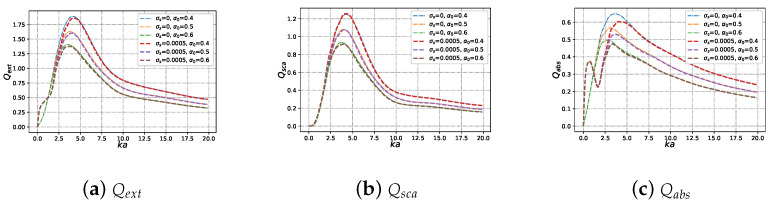
The same as Figure 9 but with different beam scaling parameter $\alpha_{0}$ values ($\alpha_{0}=\left[0.4,\;0.5,\;0.6\right]$).

**Figure 11 micromachines-15-00975-f011:**
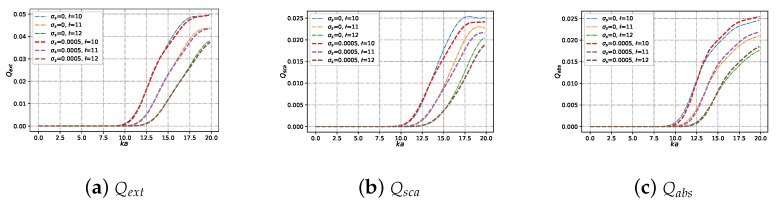
The same as Figure 9 but with different beam order ł (ł=[10,11,12]).

**Figure 12 micromachines-15-00975-f012:**
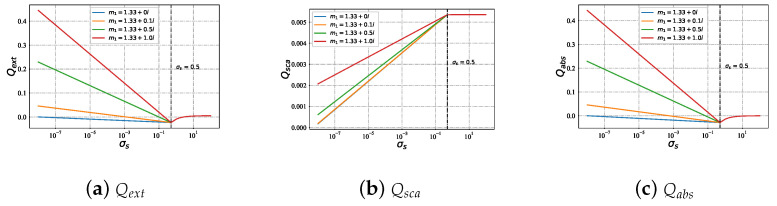
$Q_{ext}$, $Q_{sca}$, and $Q_{abs}$ under the Bessel pincer light-sheet beam on a charged particle (with different imaginary $n_{p}$ of refractive index) varying with the varying surface charge $\sigma_{s}$. The refractive index of particle is $m_{1}$ = $1.33+10^{-6}i$, the beam scaling parameter is $\alpha_{0}=0.2$, and the beam order is ł=10.

**Figure 13 micromachines-15-00975-f013:**
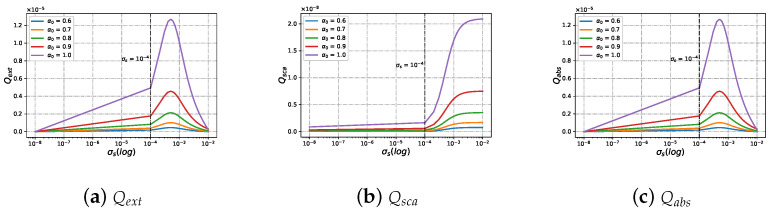
$Q_{ext}$, $Q_{sca}$m and $Q_{abs}$ under different beam scaling parameter $\alpha_{0}$ values of a Bessel pincer light-sheet beam on a charged particle varying with the varying Sigma $\sigma_{s}$. The refractive index of particle is $m_{1}$ = $1.33+10^{-6}i$, and the beam order is ł = 10.

**Figure 14 micromachines-15-00975-f014:**
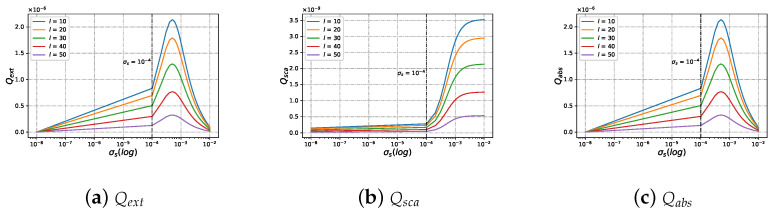
$Q_{ext}$, $Q_{sca}$, and $Q_{abs}$ under different beam order ł of a Bessel pincer light-sheet beam on a charged particle varying with the varying Sigma $\sigma_{s}$. The refractive index of particle is $m_{1}$ = $1.33+10^{-6}i$, and the beam scaling parameter is $\alpha_{0}=$ 0.8.

**Figure 15 micromachines-15-00975-f015:**
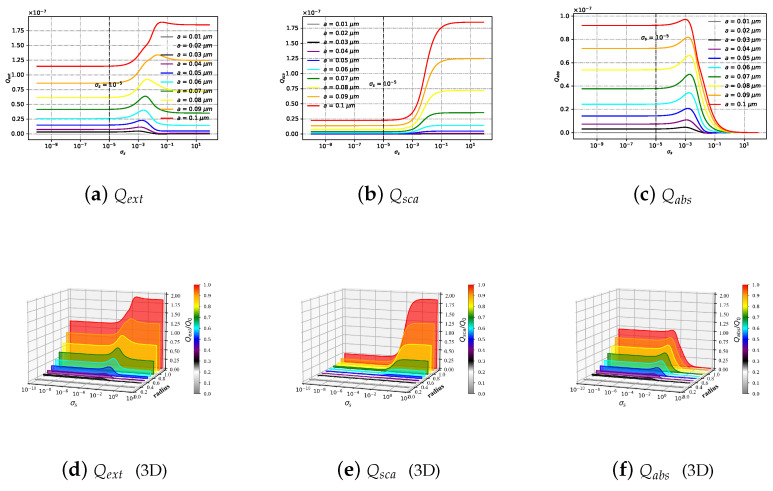
$Q_{ext}$, $Q_{sca}$ and $Q_{abs}$ of a Bessel pincer light-sheet beam on a charged particle with different particle radius (radius = *a*), varying with the varying surface charge $\sigma_{s}$. The refractive index of particle is $m_{1}$ = $1.33+10^{-6}i$, and the beam scaling parameter is $\alpha_{0}=$ 0.8.

## Data Availability

Some or all data, models, or code generated or used during the study are available in a repository (https://github.com/ShaeZhang/Datass.git, accessed on 2 July 2024).

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
