# Peer review of "Scattering of a Bessel Pincer Light-Sheet Beam on a Charged Particle at Arbitrary Size"

_micromachines, 2024, doi:10.3390/mi15080975_

Round 1
Reviewer 1 Report
Comments and Suggestions for Authors
In the paper, the authors present numerical simulation of the Bessel pincer light-sheet beam scattering on the charged spherical particle of arbitrary size. The manuscript is written clearly and, in my opinion, will be of interest to researchers in this field. Although the problem under consideration is not new, at the same time, these studies are of certain practical interest and the manuscript, after some modification, can be accepted for publication.
I have a few comments:
1.The main purpose of the manuscript and its impact should be presented more clearly in the Introduction.
2.In my opinion, the use in Figs. 2-8 of both polar and Cartesian coordinate systems, to represent the angular dependence of the far-field scattering intensity, does not provide additional important information, but only overloads the article and may confuse readers. I recommend that authors use only one coordinate system and remove unnecessary dependencies.
Reviewer 2 Report
Comments and Suggestions for Authors
Comments on micromachines-3112580: Scattering of Bessel pincer light-sheet beam on the charged particle at arbitrary size, by Shu Zhang et al.
The abstract clearly states the research objectives, focusing on the scattering characteristics of charged particles under Bessel pincer light-sheet beams, which are well-defined and relevant to electromagnetic scattering and optical manipulation. The introduction thoroughly reviews previous studies, effectively setting the context for the current research. It references pertinent many works, indicating a deep understanding of the topic. The methods section is detailed, explaining the mathematical models and assumptions made. The results section presents extensive data on scattering intensities, efficiencies, and the effects of various parameters (e.g., particle size, wavelength, surface charge). Using both polar and Cartesian plots helps visualize the scattering patterns effectively. The results are very interesting. However, there are still some issues to be addressed:
1、Add a sentence like, "This study reveals for the first time the significant differences in scattering characteristics of Bessel beams on charged particles compared to traditional beams." to briefly mention the main innovative points of this paper in the abstract, such as new findings regarding the interaction between Bessel beams and charged particles.
2. Simplify complex sentences to make the authors feel their work is more accessible. For instance, splitting the sentence into two and correcting 'multiple expanding' to 'multiple expansions' in the third sentence of the abstract will improve the paper's readability and make the authors feel their work is more easily understood.
3. Enhance the flow and readability of the text by combining the clauses with 'such as' in the first sentence of the introduction. This will make the authors feel their work is more engaging and exciting.
4、Consistency in Terminology: Ensure consistent use of terminology throughout the paper. For instance, terms like "Bessel pincer light-sheet beam" should be used uniformly to avoid confusion.
5、Verify that all references are correctly formatted and up-to-date. Including more recent studies if they are relevant might also be beneficial.
Comments on the Quality of English LanguageEnhance the flow and readability of the manuscript by polish the Engligh.
Reviewer 3 Report
Comments and Suggestions for Authors
In this theoretical manuscript Shu Zhang et. al. study the interaction of a Bessel beam with a charged sphere. Optical scattering by charged spheres has already been considered in the past, but the configuration of optical field, namely Bessel beam, seems to be new.
The topic is interesting scientifically, and authors named some applications of this configuration. The results are scientifically sound. However, certain points need to be improved prior to publication.
1. The authors mentioned several times "optical manipulation" and "optical methods", but they focus on the THz frequency range. It would be good to mention how terahertz beams are generated, and why THz is preferred over visible light as an application of the theory.
2. There are inconsistent notations of the beam order, cf. lines 105 and 110. I find the notation in line 105 is not conventional.
3. Eq.(1) needs more explanations. For instance, it is not clear what is the meaning of indices 1 and 2, or how Eq.(4.1) in Ref.[47] was split into 2 equations (1.1) and (1.5) in the present work.
4. It is not clear how the electrostatic potential (lines 151, 290) was included in the calculations.
Apart from these important points the authors should check spelling or form of the equations at the following lines: 88, 94, 124, 137, 149 (sphere ->spherical), 225, 227 (charged ->charge), 309, 292, 305, 316, 327, 338, 346, 354(Q in math mode), 331, 335, 361.
Comments on the Quality of English LanguageThe quality of English is acceptable, but there are some misprints that need to be corrected.
